# Corrosion Behavior of Tubing in High-Salinity Formation Water Environment Containing H₂S/CO₂ in Yingzhong Block

Xuehui Zhao [1],*, Junlin Liu [2], Baisheng Yao [3], Cheng Li [2], Xue Xia [2] and Anqing Fu [1]

1 State Key Laboratory for Performance and Structure Safety of Petroleum Tubular Goods and Equipment Materials, CNPC Tubular Goods Research Institute, Xi'an 710077, China
2 Research Institute of Drilling and Production Technology, PetroChina Qinghai Oilfield Company, Dunhuang 736202, China
3 The No.1 Oil Production Plant, Changqing Oilfield Company of Petrochina Co., Ltd., Yanan 716000, China
* Correspondence: zhaoxuehui@cnpc.com.cn

**Abstract:** To clarify the corrosion behavior of P110SS material under the synergistic action of multiple factors such as a $CO_2/H_2S$ coexistence environment, a high temperature, and high-salinity formation water, a series of simulation tests and analyses were carried out in this paper. High-temperature high-pressure autoclaves, scanning electron microscopy, and a three-dimensional microscope were used to analyze and evaluate the changing trend of the corrosion performance of P110SS tubing material under different temperatures and a $H_2S/CO_2$ partial pressure ratio in a high-salinity formation water environment, and the corrosion cracking sensitivity and pitting sensitivity of the material with stress were compared and analyzed. The results indicate that the average corrosion rate of P110SS material without stress increases with the rising test temperature, and the corrosion damage worsens with an increase in the $H_2S$: $CO_2$ partial pressure ratio. The highest corrosion rate for P110SS material is 0.99 mm/a. When the test temperature varies from 80 °C to 180 °C and $P_{H_2S}$:$P_{CO_2}$ = 0.53:0.17, the P110SS material with a loading stress of 85% $YS_{min}$ is not susceptible to stress corrosion cracking (SCC). Although surface pitting nucleation is evident at a high temperature of 180 °C, no expansion-induced cracking or fracture phenomena occur.

**Keywords:** $CO_2/H_2S$ corrosion; high salinity; stress corrosion sensitivity; pitting; synergistic effect





## 1. Introduction

As the unique channel for oil and gas well production, the oil casing string is essential for oil and gas exploration, development, and safe production. At present, with China's continuous expansion in the field of oil and gas exploration and development, the service environment for oil casing strings has become increasingly complex and harsh [1,2]. Especially in deep and complex exploration and development conditions, the pipe string is not only subjected to tension, pressure, bending, torsion, and other combined stress loads, but also suffers from the synergistic effect of extreme corrosion factors such as high temperature, high pressure, high salinity, and $CO_2/H_2S$, which poses severe challenges to the long-term safe and reliable service of the pipe string as well as the efficient development of oil and gas resources [3]. The operating environment of high-temperature and high-pressure (HTHP) oil and gas wells in the Yingzhong Block of the Qinghai Oilfield is relatively harsh, mainly manifested in the following "four high characteristics": deep reservoir burial (≥6000 m), high formation temperature (≥190 °C), high formation pressure (≥110.9 MPa), and high fluid salinity (≥290 g/L). The reservoir contains both $H_2S$ and $CO_2$ corrosive gases, and the corrosive environment is extremely complex. In particular, the concentration of $Cl^-$ in formation water is as high as 186 g/L, almost reaching the brine saturation point. This rare and highly saline and corrosive medium poses a severe challenge for material selection and string application optimization both domestically and internationally.

At present, a lot of research is being conducted on the corrosion law and mechanism of oilfield-produced water and the mineralization degree at home and abroad [4–6], mainly reflected in an environment with a low temperature, a $Cl^-$ content $\leq$ 60 g/L, and a salinity not exceeding 200 g/L. Wang Shutao et al. [7] investigated the sulfide stress corrosion cracking susceptibility of P110SS casing material under conditions of a high $H_2S/CO_2$ content in formation water with a salinity of 67 g/L in the Puguang gas field, and found that the material has a high stress corrosion cracking sensitivity at a low wellhead temperature of 50 °C. Deng Hu et al. [8] studied the environmental cracking behavior of C110 casing at a low temperature of 60 °C in oil field formation water containing $H_2S/CO_2$, and compared and analyzed the stress sensitivity of the material under different conditions, and the results show that the stress sensitivity of the C110 material was relatively high, and the surface W-Ni coating and heat treatment could effectively reduce the environmental cracking sensitivity. Huang Shilin et al. [9] studied the factors affecting the corrosion of P110SS steel in the simulated environment of annular fluid in a sulfur-bearing gas well. The solution medium was prepared using NaCl and $Na_2S$, and the results show that the main factors affecting the corrosion rate of P110SS steel were temperature, followed by PH value and chloride ion content. Liu Junlin et al. [10] studied the corrosion behavior of P110/P110SS oil casing string in a high-salinity annular protective fluid environment, indicating that the corrosion rate of materials in an environment containing a small amount of dissolved oxygen was relatively increased, and the material was not sensitive to stress corrosion cracking at a high temperature of 180 °C. Li Mingxing et al. [11] studied the corrosion law of tubing in the 80S of formation water with a $Cl^-$ concentration of 36.6 g/L and containing $H_2S/CO_2$, and determined that when the $H_2S$ concentration was below the range of 0.1 MPa, the material corrosion rate decreased with the increase in the $H_2S$ concentration, and the corrosion rate showed an upward trend with the increase in temperature. Wang Yunfan et al. [12] studied the corrosion law of P110SS steel with high $H_2S$ and $CO_2$ contents in a formation water environment with a salinity of 67 g/L, and showed that with the increase in the $H_2S$, $CO_2$ partial pressure, and temperature, the corrosion rate of the P110SS steel first decreased and then increased. When the bottom hole temperature is 130 °C, the SSCC sensitivity of the material decreases, and the temperature plays a major role. Therefore, the produced formation water environment in the Yingzhong Block is relatively harsh. In a relatively high-salinity environment, the oil casing string has a serious corrosion failure risk under the synergistic action of multiple factors such as high temperature corrosion, tensile stress, $CO_2/H_2S$ and scale. In particular, the stress corrosion cracking sensitivity of the string under an $H_2S$ environment and a high-temperature environment is of great importance [13,14].

This paper is based on the safety requirements of pipe string service; it is more important to actively study the corrosion performance of oil casing string in a field environment containing $H_2S/CO_2$ and the coupling effect of multiple factors in the oilfield. With the increase in the well depth and the increase in the pipe string's service temperature, it is especially necessary to clarify the influence of high-salinity formation water combined with $H_2S$, high temperature, and other factors on the corrosion behavior of pipe string. This paper provides a reliable theoretical basis and technical support for the optimization and selection of test production wells in Yingzhong Block.

## 2. Experimental Procedure

### 2.1. Material and Solution

The material used for the test was a commercial P110SS tubing extracted from the oil field, of which the chemical composition (by mass fraction) was 0.20% C, 0.21% Si, 0.51% Mn, 0.001 3% S, 0.009 2% P, 0.52% Cr, 0.03% Ni, 0.73% Mo, and 0.046% Cu, and the remaining part was composed of Fe.

There were two types of samples, no-stress state and applied stress state. Specifically, the size of specimens was 40 mm × 10 mm × 3 mm, with a Φ 6 mm hole on one end. The stressed samples were 115 mm × 15 mm × 3 mm in size, in which the stress, equal to 85%

of the nominal yield strength ($YS_{min}$) of the material, was applied using the four-point bending method, and stress loading was carried out according to the loading mode and stress calculation method of method E in standard GB/T 4157-2017. Before the test, all the samples were carried out in accordance with the treatment procedures specified in the standard GB/T 15970.2-2000 (ISO 7539-2) so as to avoid and reduce the existence of residual stress. After polishing, the samples were cleaned with distilled water and ethanol, dried under cool air, and stored in a dry $N_2$ atmosphere [15]. The high-salinity formation water extracted from the oilfield site was adopted as the test solution, and the ion concentration of the water sample was analyzed, as shown in Table 1. Before the test, nitrogen was fed into the solution for deoxygenation treatment, and nitrogen was continuously connected to the autoclave after the sample was installed to remove the oxygen. Then, the solution was heated to the required temperature, and the $CO_2$ and $H_2S$ gases were added to the required partial pressure values. The test parameters are shown in Table 2, where $H_2S$ partial pressure ($P_{H_2S}$) is divided into two types, $P_{H_2S} = 0.35$ MPa and $P_{H_2S} = 0.28$ MPa, which were to simulate the oil field's environment with different $H_2S$ and $CO_2$ partial pressure ratios. Based on the actual situation of the oil field, the produced fluid comprises three phases of oil, gas, and water. As the produced fluid flows through the tubing, it may come into contact with either the gas phase or formation water (no consideration was given to the contact with crude oil to slow down corrosion here). Therefore, it is necessary to consider installing the sample in a gas and liquid phase environment for evaluation during the simulation testing and to control the amount of solution added according to the volume of the autoclave during the test. The schematic diagram of sample installation is shown in Figure 1.

**Table 1.** Analysis of ion composition of field formation water (g/L).

| $Cl^-$ | $Mg^{2+}$ | $Ca^{2+}$ | $K^+$ | $Na^+$ | $SO_4^{2+}$ | $HCO_3^-$ | PH | Salinity |
|---|---|---|---|---|---|---|---|---|
| 186 | 0.035 | 0.11 | 7.05 | 104 | 24.3 | 1.07 | 6.28 | 292 |

**Table 2.** Simulation test parameters.

| Items | Test Temperature (°C) | $H_2S$ Content (MPa) | $CO_2$ Content (MPa) | Total Pressure (MPa) | Test Period (h) |
|---|---|---|---|---|---|
| Parameters | 80<br>120<br>180 | 0.53<br>0.28 | 0.17 | 10 | 168 |

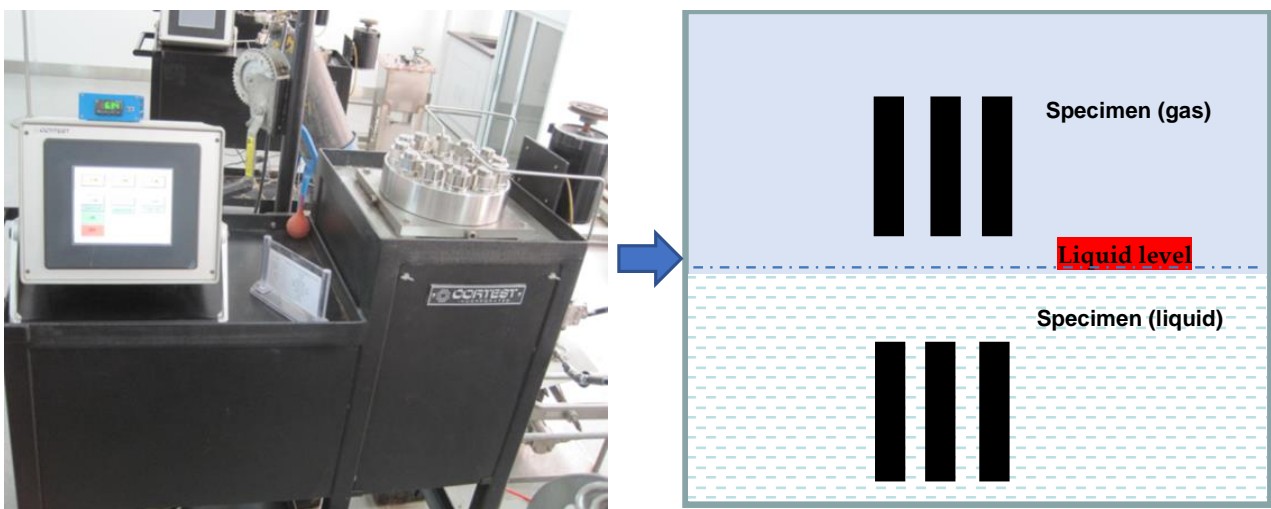

**Figure 1.** Installation diagram of samples in autoclave.

### 2.2. Weight Loss Tests

Weight loss tests were conducted in an autoclave to investigate corrosion rate in oilfield formation water. Prior to the weight loss tests, the samples were cleaned with distilled water and acetone, dried, and then weighed using a balance with a precision of 0.1 mg. The weight values were recorded as the original weight ($W_{0i}$, $i = 1, 2, \ldots$). After the tests, all samples were taken out and immediately cleaned using distilled water and acetone. Then, the corrosion product scale on the sample's surface was removed with the film removal solution at room temperature and then rinsed and dried. After that, the samples were weighed again to obtain the final weight. The corrosion rate ($V_{corr}$) was calculated in mm/y (average corrosion thinning depth in years) from the weight loss by using Equation (1) [16] as follows:

$$V_{corr} = 8.76 \times 10^6 \times (W_{0i} - W_{1i})/(t \times \rho \times S) \ (i = 1, 2, \ldots) \tag{1}$$

where $W_{0i}$ and $W_{1i}$ are, respectively, the original and final weights of the samples in g, S is the exposed surface area of the samples in mm$^2$, t represents the immersion time in h, and $\rho$ is the steel density equaling $7.8 \times 10^{-3}$ g·mm$^{-3}$. An average corrosion rate of the three different samples for each test condition was reported as an overall corrosion rate for each set of conditions.

### 2.3. SCC Testing

Stress corrosion cracking testing was carried out using the immersion method under high temperature and high pressure conditions in an autoclave. In these tests, all sample surfaces required precision machining and a surface roughness of Ra $\leq$ 0.2 $\mu$m. The stress (85% $YS_{min}$) was applied using the four-point bending method according to the GB/T 4157-2017, and stress loading can be achieved by Equation (2) as follows:

$$\sigma = 12E \times t \times y/(3H^2 - 4A^2) \tag{2}$$

where H represents the distance between the two outermost fulcrums, E is the elastic modulus, A represents the distance between the inner and outer fulcrums, t is the sample thickness, and $\sigma$ represents the stress value of the loading. Using these known parameters to calculate the maximum deflection y between the outer fulcrums, the stress is loaded by detecting the deflection change of the specimen. The schematic diagram of the four-point bending loading device is shown in Figure 2 [17].

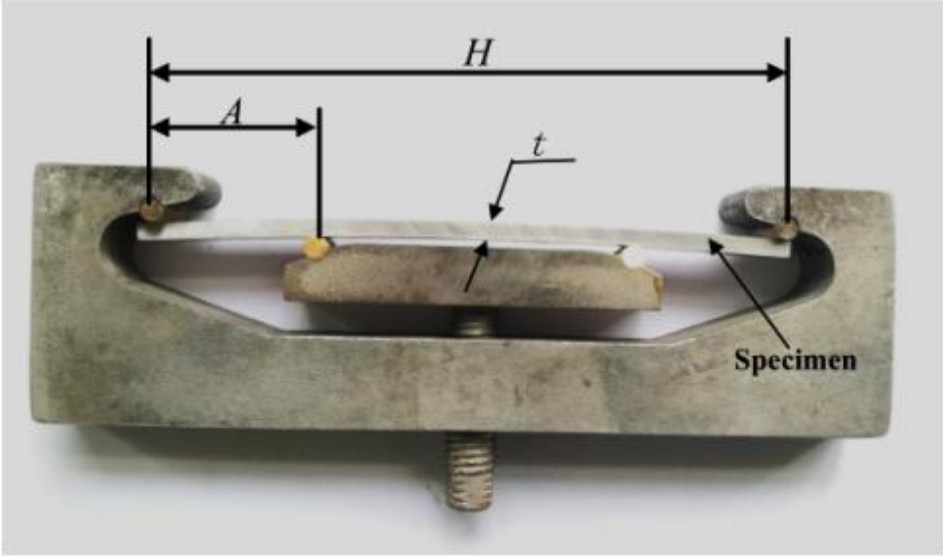

**Figure 2.** Schematic diagram of stress application using four-point bending method.

### 2.4. Characterization

The corrosion products were studied using scanning electron microscopy (SEM) and optical microscopy (OM). The elemental composition of the corrosion products was quantitatively analyzed using EDS spectrometer. For the samples under stress, it is necessary to observe any crack morphology and corrosion microstructure.

## 3. Results and Analysis

### 3.1. Average Corrosion Rate

After the test, the average corrosion rate of the material under two test conditions at different temperatures was calculated using the weight loss method. Figure 3 shows the trend of the average corrosion rates ($V_{corr}$) of the material. It can be seen from Figure 3 that the corrosion rate of the material under $P_{H_2S}$ = 0.53 MPa is significantly higher than that under $P_{H_2S}$ = 0.28 MPa when the test temperature is $\geq$120 °C, indicating that increasing the $H_2S$ partial pressure under the same condition can intensify the material corrosion. As the test temperature increases, the average corrosion rate of the material in the liquid phase environment gradually increases. The corrosion level of the material was evaluated according to NACE SP0775-2023. Thus, it is indicated that the material exhibits low corrosion at the test temperature of 80 °C, moderate corrosion at 120 °C, and high corrosion at 180 °C. Therefore, when the service temperature of the P110SS pipe is $\geq$120 °C, it is recommended to take anti-corrosion measures, such as adding a matching corrosion inhibitor or coating, to prevent or slow down the corrosion damage of the pipe.

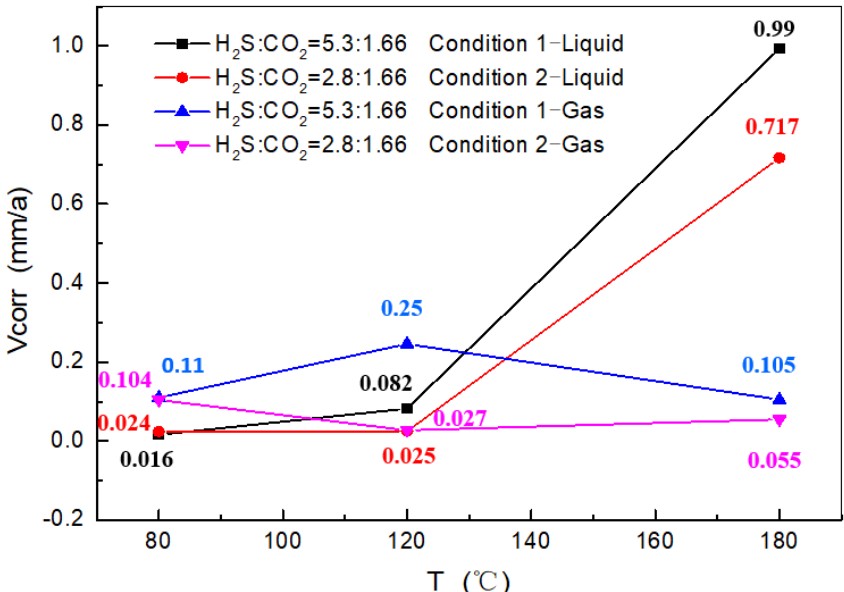

**Figure 3.** Variation trend of corrosion rate of P110SS tubing material under different temperature environments.

By comparison, it can be learned that there is no significant change in the corrosion of materials in a gas phase environment with the temperature. This relates to the extent of random adsorption and condensation of moisture on a material's surface in a gas phase environment. When the temperature is $\leq$120 °C, the materials in a gas phase environment are more susceptible to corrosion than those in a liquid phase environment. However, the corrosion of materials in a liquid phase environment is significantly higher than that in a gas phase environment at a temperature of 180 °C, indicating that the effect of a high temperature increases the surface activity of the material and accelerates the electron transfer between the material and the solution medium, thereby rapidly strengthening the electrochemical interaction of the material with $CO_2$ and $H_2S$ dissolved in water, further speeding up the corrosion [18–20].

### 3.2. Corrosion Morphology Characteristics

3.2.1. Macroscopic Corrosion Morphology under $P_{H_2S}$ = 0.53 MPa

Figure 4 shows the macroscopic corrosion morphology of P110SS material under a simulated high-pressure gas phase environment of oilfield formation water at various temperature conditions. It can be observed that the sample's surface loses its metallic luster and is relatively rough, presenting an uneven spot corrosion. Due to the varying degrees of condensation of moisture containing corrosive gases on the surface of the sample in a gas phase environment, the corrosion varies in different areas. In particular, the local corrosion caused by moisture condensation is more obvious at the high temperature of 180 °C, resulting in relatively large differences in the spotted color on the surface. By comparing the macroscopic corrosion morphology of P110SS materials in a liquid phase environment under the same condition (Figure 5), it can be seen that the surface of the sample is greyish black, flat, and uniform in general, but it is uniform and significantly rough at the high temperature of 180 °C, without obvious pitting corrosion. This indicates that the surface of the material in the liquid phase environment has undergone uniform corrosion due to the sufficient contact with the corrosive media.

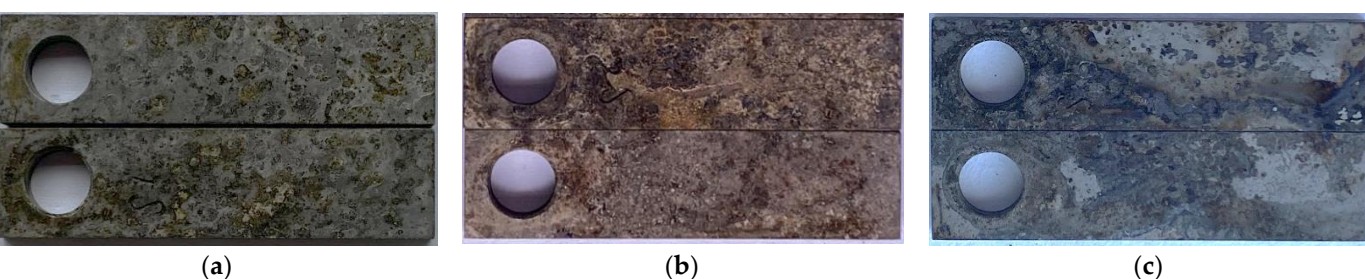

| (a) | (b) | (c) |

**Figure 4.** Macroscopic corrosion morphology of materials in gas environments under different temperatures; (**a**) 80 °C, (**b**) 120 °C, (**c**) 180 °C.

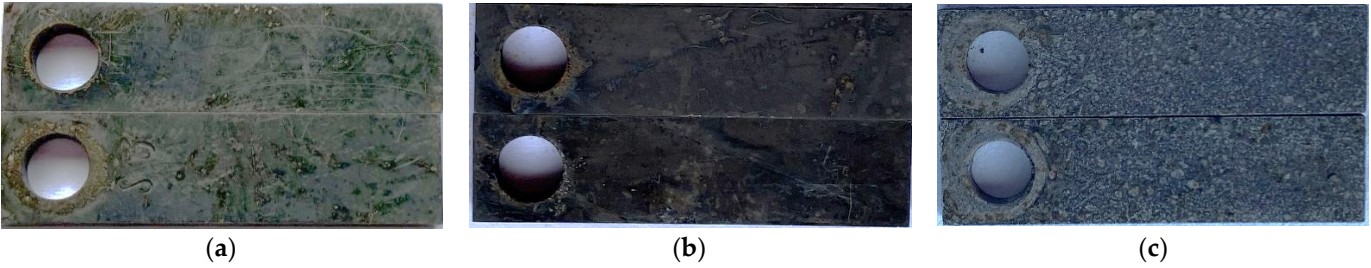

| (a) | (b) | (c) |

**Figure 5.** Macroscopic corrosion morphology of materials in liquid environments under different temperatures; (**a**) 80 °C, (**b**) 120 °C, (**c**) 180 °C.

3.2.2. Micro-Corrosion Morphology When $P_{H_2S}$ = 0.53 MPa

A scanning electron microscope was used to observe the microscopic corrosion morphology of P110SS material in gas and liquid phase environments. Figure 6 displays the corrosion morphology of the material under gas phase conditions at different temperatures, where uneven and rough corrosion product films are clearly visible, and there is variation in the accumulation of products across different regions. This indicates that the moisture is adsorbed and condenses differently on the surface of the sample, with droplets condensed locally, causing obvious local corrosion, and with thin infiltration films in some areas, resulting in varying corrosion. Figure 7 shows the microscopic corrosion morphology of P110SS material in a liquid phase environment, from which it can be seen that the surface of the specimen is relatively flat at the temperature ≤ 120 °C compared to that in the gas phase environment. The material is uniformly exposed to a corrosive medium in an environment of solution immersion, with an equal probability of corrosion occurring. The corrosion product film peels off locally on the surface. At the high temperature of 180 °C,

the corrosion product film is especially rough and relatively thick, with the surface film partially peeled off, so that the bottom layer of the corrosion product film can be observed, without obvious pitting corrosion.

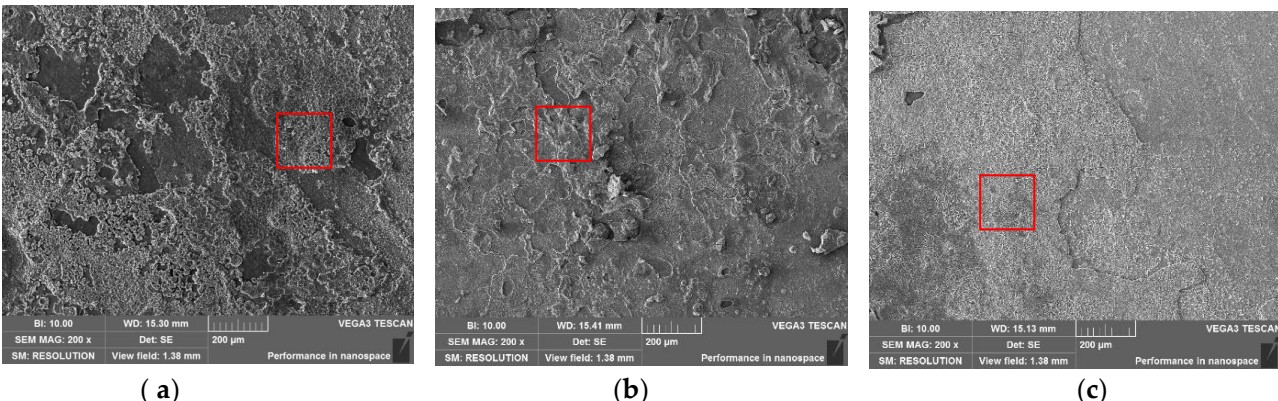

**Figure 6.** Microscopic corrosion morphology of materials in gas environments under different temperatures; (**a**) 80 °C, (**b**) 120 °C, (**c**) 180 °C.

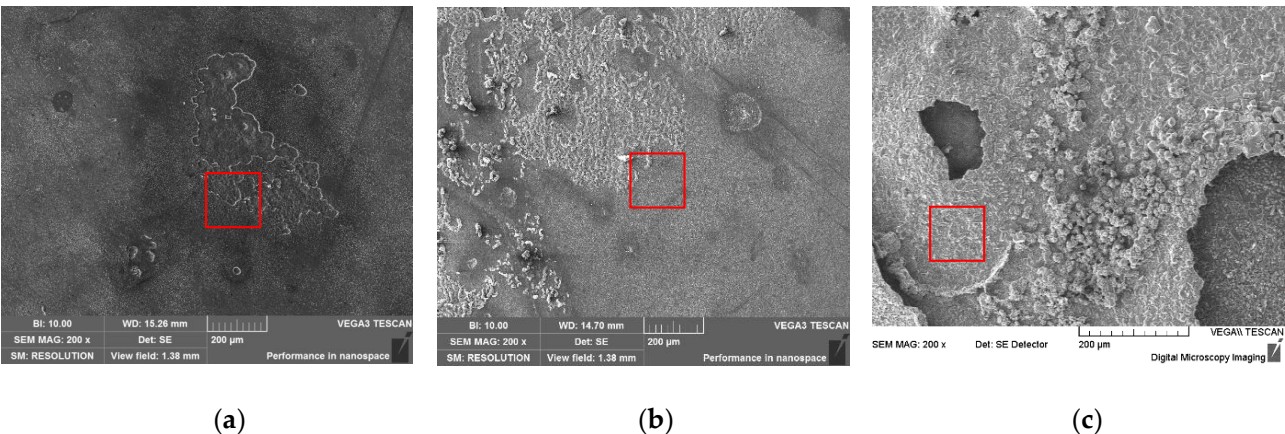

**Figure 7.** Microscopic corrosion morphology of materials in liquid environments under different temperatures; (**a**) 80 °C, (**b**) 120 °C, (**c**) 180 °C.

EDS was used to analyze the elemental composition of the corrosion product film on the surface of the material (see the red box area in Figures 6 and 7). The results show that the corrosion product film on the surface of the gas phase and liquid phase are mainly composed of O, S, and Fe elements (Table 3), and with the increasing test temperature, the content of S in the corrosion product film increases gradually. However, no calcium and magnesium ions that are prone to scaling were found, so it was concluded that the corrosion product mainly consisted of a mixture of FeS and $FeCO_3$.

**Table 3.** EDS analysis results of product films on the surfaces of materials under different test conditions.

| Ambient Temperature Elements | 80 °C | | 120 °C | | 180 °C | |
|---|---|---|---|---|---|---|
| | Gas | Liquid | Gas | Liquid | Gas | Liquid |
| O $W_t$% | 7.12 | 8.82 | 8.69 | 9.86 | 9.29 | 8.48 |
| S $W_t$% | 29.09 | 10.21 | 32.49 | 18.64 | 29.41 | 32.47 |
| Fe $W_t$% | 63.79 | 80.09 | 58.82 | 73.49 | 59.77 | 59.06 |

By comparing the element contents of the surface corrosion products under the liquid immersion environment, it can be seen that the content of S shows an increasing trend with the gradual increase in the temperature, indicating that the high-temperature environment promotes the intensification of $H_2S$ and substrate corrosion, and that the FeS content in the corrosion film increases, leading to more severe corrosion [21,22].

Figure 8 shows the cross-sectional morphology of the product film at the high temperature of 180 °C, where the product film is in greyish black. EDS analysis was performed on the corrosion products in the red box. The result shows that the film is mainly composed of O, S, and Fe elements. As shown on the cross section, the substrate under the film is uneven due to corrosion, indicating that corrosion will further grow to the deeper layers when the corrosion product film on the surface is unable to resist the penetration of the corrosive media. The corrosion morphology indicates that the material constantly changes in the dynamic balance of " corrosion product film formation-film peeled off-film formation". The formation and densification of the film can slow down the further growth of the corrosion. On the contrary, when the film is loosened, a similar form of under-scale corrosion will occur [23–27].

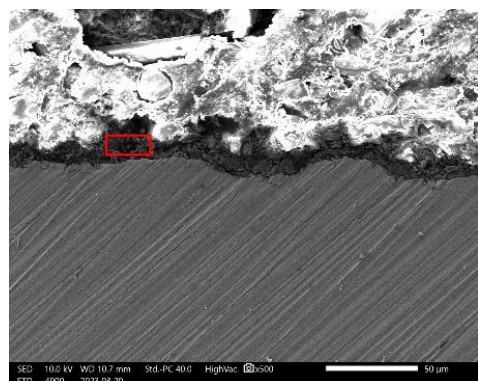 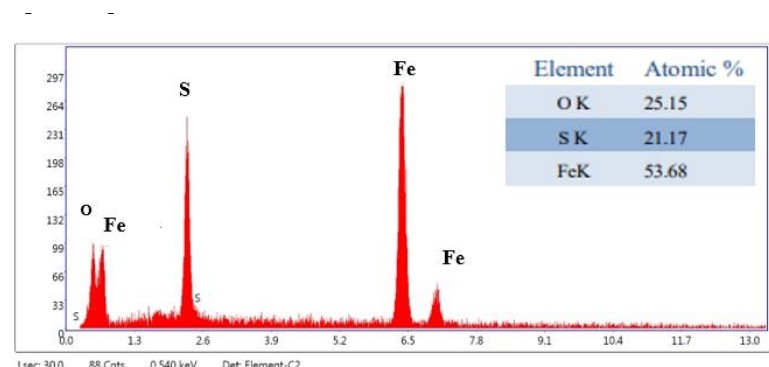

**Figure 8.** Morphology and energy spectrum analysis on cross section of product film under ultra-high temperature of 180 °C.

### 3.2.3. Microscopic Corrosion Morphology under $P_{H_2S}$ = 0.28 MPa

Based on the change in the partial pressure of the corrosive gases in the produced fluid at the oil field, the corrosion performances of the materials at the decreasing $H_2S$ partial pressure were analyzed comparatively. Figure 9 shows the surface corrosion morphology under different gas phase conditions at different temperatures. By comparison, it can be seen that the different areas on the surface are in different states due to the adsorption and condensation of moisture on the specimen surface. Generally, the spotted morphology is presented due to the accumulation of surface product, with local roughness as a result of corrosion. Figure 10 shows the micro-corrosion morphology of the sample in a liquid phase environment with $P_{H_2S}$ = 0.28 MPa. Compared with $P_{H_2S}$ = 0.53 MPa, the surface of the sample is relatively flat, especially when the simulated temperature is ≤120 °C, when the surface of the material is flat and uniform, and when there is no obvious local corrosion phenomenon. It can be seen that the corrosion degree is relatively reduced from the morphology when the $H_2S$ content is relatively reduced, which is consistent with the change trend of the corrosion rate above. It shows that the presence and concentration of $H_2S$ have a great influence on the erosivity of solution system, which is of great significance for the adaptability evaluation of anti-sulfur materials (such as P110SS) and the definition of critical indicators in an application environment. Similarly, the relatively thick corrosion product film is visible at a high temperature of 180 °C, and a surface film and a bottom product film form after the surface film peels off. Hence, the corrosion is still severe in high-temperature environments, and the film's roughness is relatively reduced compared to that in the high-concentration $H_2S$ environment with $P_{H_2S}$ = 0.53 MPa.

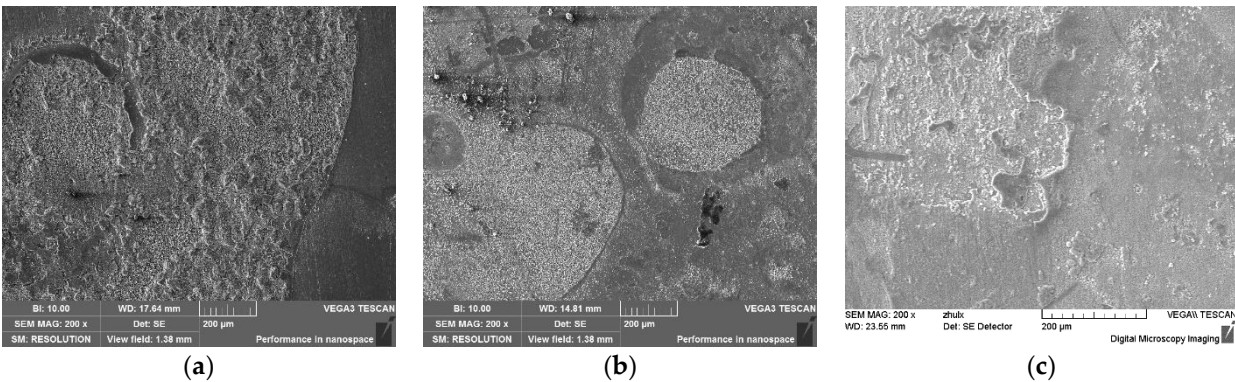

**Figure 9.** Microscopic corrosion morphology of materials under gas phase environments at different temperatures; (**a**) 80 °C, (**b**) 120 °C, (**c**) 180 °C.

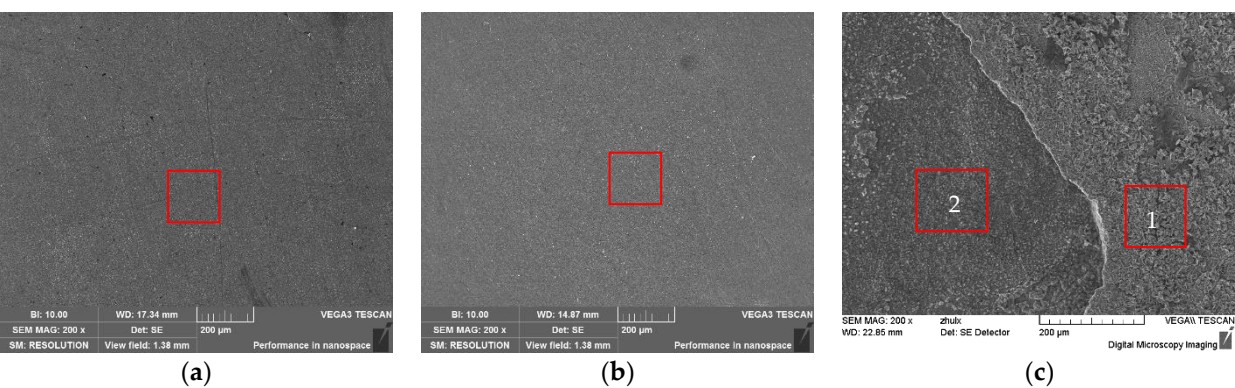

**Figure 10.** Microscopic corrosion morphology of materials under the liquid phase environments at different temperatures; (**a**) 80 °C, (**b**) 120 °C, (**c**) 180 °C.

EDS was used to analyze the corrosion product films on the surfaces of the materials under liquid phase conditions at different temperatures (Figure 11), and it was found that they were mainly composed of O, S, and Fe elements. At the same time, with the increase in the simulated test temperature, the product films were relatively thicker at 180 °C, and a second product film was visible at the shed area, indicating that the bare substrate continued to be corroded, and the content of S in the product films gradually increased. These results indicate that in the co-existence environment of $CO_2/H_2S$, corrosive gases cooperate with the high temperature to accelerate the electrochemical action of the solution system and promote the intensification of corrosion. Although the increasing temperature decreases the solubility of corrosion gas, the activity of the solution system increases with the increasing temperature, accelerating electron transfer. The $Cl^-$ ion in the product film is the residue of the solution medium.

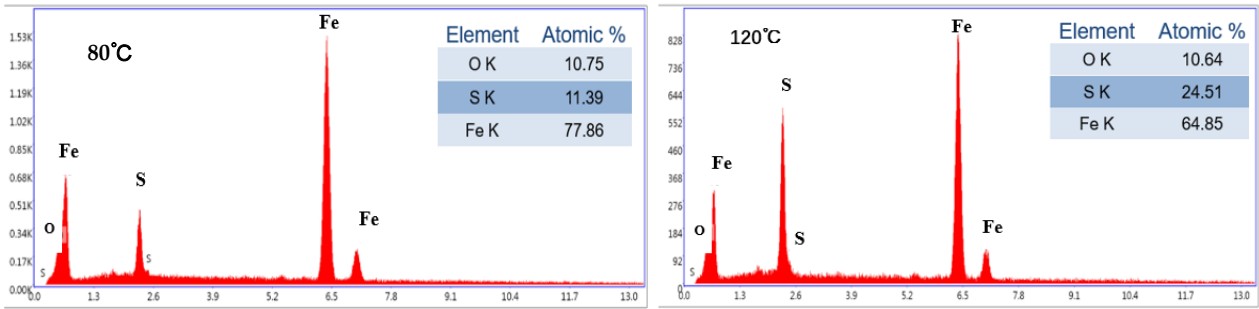

**Figure 11.** *Cont.*

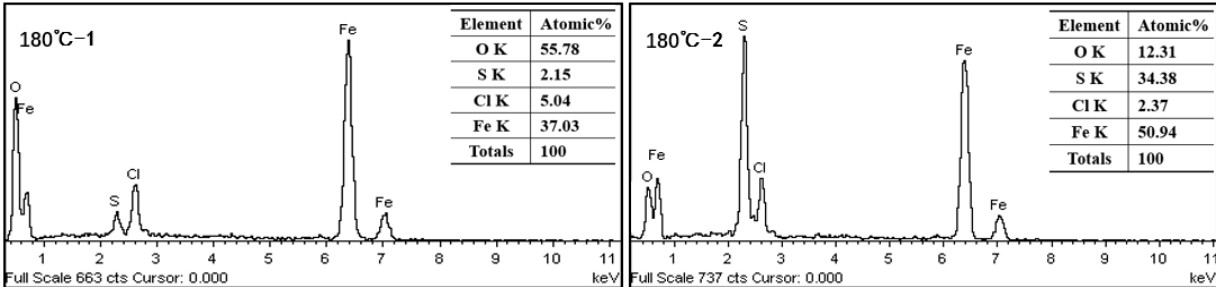

**Figure 11.** EDS analysis results of product films on the surfaces of materials under the liquid phase environments.

### 3.3. SCC in H₂S-Containing Environment

　　In order to study the stress corrosion cracking sensitivity of P110SS material in the high-salinity and high-temperature environments containing $H_2S$, the test was conducted by simulating the formation water environment using the high-temperature and high-pressure device. The test parameters are shown in Table 4.

**Table 4.** Specific test parameters (total pressure = 10 MPa).

| Items | Temperature/°C | H₂S Content (MPa) | CO₂ Content (MPa) | Test Cycle/h | State | Applied Stress |
|---|---|---|---|---|---|---|
| Parameters | 80, 120, 180 | 0.53 | 0.17 | 720 | Liquid | $85\%YS_{min}$ |

　　After 720 h of testing, no cracking or fractures were found in the samples. Figure 12 shows the surface corrosion morphology of the samples at different temperatures, with obvious grey corrosion products on the surface of the samples under low temperature conditions (80 °C). However, no obvious local corrosion or cracking was found in the stress concentration area of the samples. Under the high temperature condition of 120 °C, the stress concentration area on the surface of the sample remains flat, but visible uniform micro-pitted morphology can be observed. This indicates that the synergistic effect of the corrosive media and stress causes the pitting nucleation on the material's surface as the temperature increases. When the temperature rises to 180 °C, the product scale in the surface stress concentration area is rough, loose, and partially detached. Additionally, it is observed by a 30× stereo microscope in the red box area of Figure 12c that the morphology of corrosion pits is present in the local detachment area (Figure 13). As observed by the stereo microscope, the depth of the pits is immeasurable, indicating the relatively shallow depth of the pits. This shows that the pitting nucleation appears on the material's surface with the increasing temperature under the condition of high-salinity formation water containing $H_2S/CO_2$ corrosive gas. In addition, the corrosion pits are relatively obvious at the high temperature of 180 °C, but the pitting does not expand to cause SCC in the specimens.

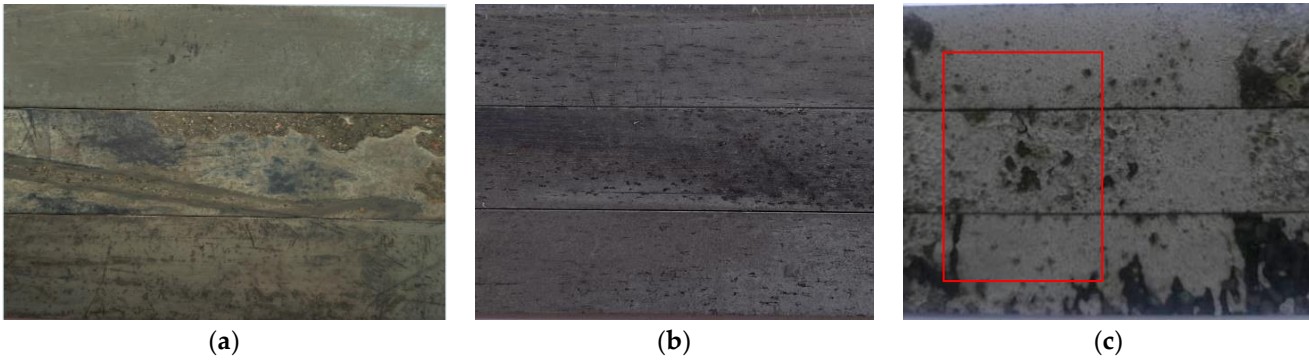

(a)　　　　　　　　　　　　　　(b)　　　　　　　　　　　　　　(c)

**Figure 12.** Stress-sensitive corrosion morphology of formation water environment at different temperatures (X30 times); (**a**) 80 °C, (**b**) 120 °C, (**c**) 180 °C.

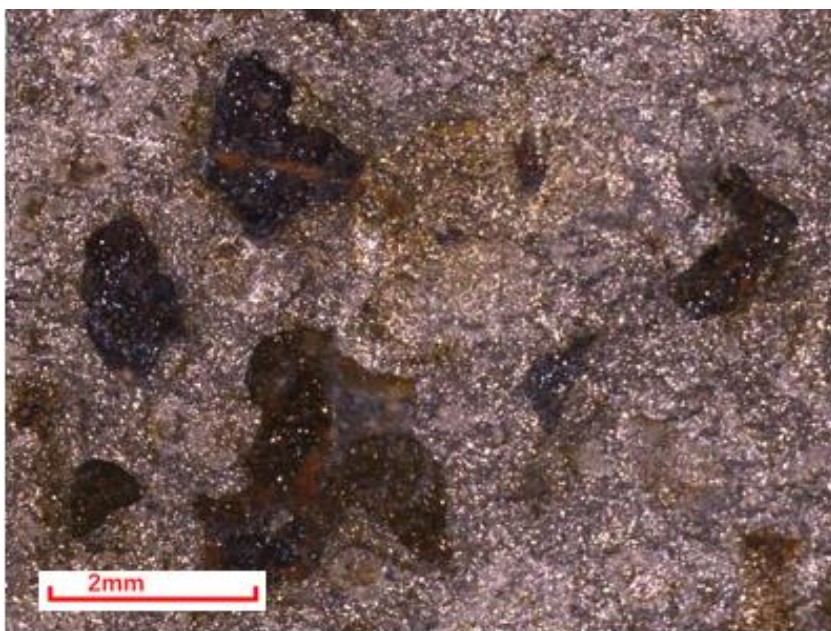

**Figure 13.** Macroscopic morphology of surface corrosion pits on the stressed specimen in 180 °C formation water environment.

*3.4. Discussion*

3.4.1. Corrosion Mechanism

In the simulated environments of high temperature and high pressure, the gas phase differs significantly from the liquid phase. When the material is in the wet environment containing $CO_2/H_2S$, some areas of the surface become soaked by the moisture and form a liquid film, resulting in relatively uniform corrosion. However, in other areas, the local liquid film gradually accumulates into droplets with high concentrations of dissolved $CO_2/H_2S$ corrosive gas. This leads to increased corrosion in those droplet areas and creates obvious local rough porphyritic corrosion morphology [28]. On the other hand, the $H_2S/CO_2$ corrosion gas dissolves and ionizes in the solution, and the ionized $H^+$ gives full play to the effect of cathode depolarization, promoting the rapid electrochemical action between the matrix material and the solution medium [29]. At the same time, the salinity in the solution medium is relatively high, which is conducive to the transfer, gain, and loss of electrons. In recent years, corrosion failures occurred frequently in high-well-depth and high-temperature environments, so the influence of temperature has been in the main position [9]. With the increasing temperature, the activity of the solution system is increased, and the surface activity of the material is strengthened, so the synergistic effects of multiple factors are promoted to deeper severe corrosion. The corrosion degree of the P110SS material increases significantly with the increasing temperature. At the high temperature of 180 °C, the corrosion product film is relatively thick, and a second film can be seen at the shedding place, indicating that the exposed matrix has a further electrochemical interaction with the solution medium rapidly after the shedding of the film layer, resulting in an increase in the corrosion rate. Therefore, in high-temperature solution systems, a dynamic equilibrium of "corrosion-film layer shedding-film layer formation" occurs on the material surface. When the corrosion product film is dense, it can alleviate further corrosion expansion; however, if the film is loose, under-scale corrosion may occur.

The EDS analysis of the corrosion product film showed no calcium–magnesium ion scaling phenomenon, indicating that although the solution medium has a high salinity, the scaling trend is weak in the test system containing $CO_2/H_2S$. In the simulated test system, the $H_2S$ and $CO_2$ corrosion gases dissolve and ionize a large amount of $H^+$, which makes the acidity of the solution increase relatively, which affects the formation of the calcium and

magnesium scale layer in the high-salinity test system. So, the corrosion failure is mainly the form of under-scale corrosion caused by the accumulation of corrosion products.

### 3.4.2. Stress Sensitivity Analysis

According to the test results on the stress cracking sensitivity of the material, the synergistic effects of high temperature and stress induce the gradual pitting nucleation of the material. As the temperature increases from 80 °C to 180 °C, the P110SS material's surface exhibits a trend of "no nucleation-small nucleation-pitting corrosion growth". Generally, the synergistic effects of high temperature and stress not only promote the intensification of corrosion, but also accelerate the Cl⁻ ions penetrating the film layer, resulting in local corrosion under the film layer. However, due to local pitting nucleation and limited deep expansion under the test environment, no cracking or fracture of the material was caused. Therefore, P110SS material is insensitive to sulfide stress corrosion cracking when $P_{H_2S}$ is 0.53 MPa at the high temperature of 180 °C.

### 4. Conclusions

(1) In the simulated formation water environment with a high salinity of 292 g/L and $H_2S/CO_2$ corrosion gas, the average corrosion rate of the P110SS material gradually increases as the temperature rises from 80 °C to 180 °C, and when $P_{H_2S}$ = 0.53 MPa and $P_{CO_2}$ = 0.17 MPa, the corrosion rate of P110SS can reach up to 0.99 mm/a.

(2) In both liquid and gas phase environments, the morphology of the corrosion product film on P110SS varies due to the different states of the material that are in contact with the corrosive medium. In a gas phase environment, a locally rough and spotty-shaped corrosion product film is formed as a result of varying degrees of aggregation in the liquid film.

(3) When the simulated test temperature gradually increases from 80 °C to 180 °C with a $P_{H_2S}$: $P_{CO_2}$ ratio of 0.53:0.17, the P110SS material shows no sensitivity to SCC when loaded with stress at 85% $YS_{min}$. However, as the temperature increases, the material becomes more susceptible to pitting corrosion. At high temperatures (180 °C), pitting occurs, but does not lead to cracking or fracturing of the material.

(4) When the simulated test temperature is ≥120 °C, the P110SS material experiences extremely severe corrosion, Therefore, it is recommended to take appropriate anti-corrosion measures when using P110SS in an environment with a temperature of ≥120 °C.

**Author Contributions:** X.Z. conducted the research and result analysis, J.L. and X.X. contributed to the design of the experiments and morphology analysis during the research activities, and B.Y., A.F. and C.L. contributed to image processing and data calculation. All authors have read and agreed to the published version of the manuscript.

**Funding:** This research was funded by the National Natural Science Foundation (52071338), the China National Petroleum Corporation for Science Research, and the technology development project (2021ZZ01-04).

**Institutional Review Board Statement:** Not appliable.

**Informed Consent Statement:** Not appliable.

**Data Availability Statement:** Not appliable.

**Conflicts of Interest:** The authors declare no conflict of interest.

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
