# Peer review of "Corrosion Behavior of Tubing in High-Salinity Formation Water Environment Containing H2S/CO2 in Yingzhong Block"

_coatings, doi:10.3390/coatings13081342_

Round 1
Reviewer 1 Report
line 69: It is unclear how the stress was measured or achieved.
71: is there a reason to use mirror finis samples in the experiments? It is quite different from what the actual metal surface used in these applications.
Table 1: The analysis is insufficient as it lacks the information about acidity and content of H2S. Use of g/l would improve legibility of the table.
Table 2: authors do not indicate type of percentage used.
Figure 1 (page 3): what method was used to assure the position of the water level?
Figure 1 (page 4 - second "figure 1") - what is the number indicated on the left of the Y axis?
line 113: NACE no longer exists, standards have been re-numbered or rebranded.
172: Unsubstantiated. For confirmation, phase analysis would have to be performed. The mixture likely also includes Fe oxohydroxides.
190: It is a feasible theory, but unsubstantiated. No proof of peeling is presented.
Table 3: EDS analysis of whole corrosion layer composition is insufficient. With layer of this thickness, EDS is likely penetrating only top portion and is therefore location-dependant.
The structure could be improved. For instance, in chap. 3.2.3 is titled "microscopic corrosion mophology under conditions 2" but it includes corrosion rates, results from condition 1 and other comparison, making it difficult to navigate.
227: EDS does not give the information about the intensification of corrosion of the medium.
229: clarification is required. The solubility of H2S decreases with temperature so the mechanism is likely different to what is described.
Figure 8: these results seem extremely unlikely. In liquid with 186 000 ppm Cl- not to detect any chlorides on the surface is simply impossible. Sample applies for Na and K. Also, quantification of oxygen using EDS is not reliable. It seems that the authors manually selected few elements and ignored all the others.
Figure 9: likely breaches copyright, looks like scan or copy from a book.
239: how was the stress calculated? Again, this is impossible to be true across the cross-section of the bent sample, the inside curve will experience much different stresses than the outside which is the reason why deformation is usually given in this experimental setup.
293: Just an opinion, but calling 180 °C "ultra-high" seems excessive.
306: if chlorides penetrate the film, they would have to be found in EDS analysis.
309: Too broad conclusion to be made based on the experiments performed.
English level is mostly ok. Nevertheless, check-up is required, there are words out-of place and non-scientific terms.
Author Response
Dear reviewer :
Thank you for the comments on the manuscript, We have revised the manuscript according to your comments. The revised manuscript and the detail responses to the comments of the editor are attached.
- line 69: It is unclear how the stress was measured or achieved.
Answer: Thanks to the reviewer's suggestion. Stress loading is carried out according to the loading mode and stress calculation method of Method E in standard GB/T4157-2017. The loading stress is 85% of the minimum yield strength of the material. A reference is added to explain how to load tensile stress and load stress value to the sample.
- 2. 71: is there a reason to use mirror finis samples in the experiments? It is quite different from what the actual metal surface used in these applications.
Answer: Thank you for your suggestions. Before the simulation test, the surface of the sample is generally polished to 800# accuracy using water abrasive paper to ensure the cleanliness and finish of the surface. This finish is close to the surface requirements of the oil pipe products used in the field, but the accuracy of the mirror can not be achieved. The author modified the statement in the article and highlighted the identification.
- Table 1: The analysis is insufficient as it lacks the information about acidity and content of H2S. Use of g/l would improve legibility of the table.
Answer: Thanks to the reviewer's suggestion. Table 1 shows the ion concentration of the solution medium during the simulation test. The author has added the PH value of the solution in the paper, and the H2S content is reflected in Table 2. The unit is changed to g/l.
- Table 2: authors do not indicate type of percentage used.
Answer: Thanks to the reminding of the reviewer. The author modified the expressions of H2S and CO2 in Table 2 and directly modified them into the partial pressure values of the two gases during the test, in MPa.
- Figure 1 (page 3): what method was used to assure the position of the water level?
Answer: Thanks to the reminding of the reviewer. The simulation test was carried out in an autoclave with a certain volume. During the test, an appropriate amount of solution was prepared according to the height of the autoclave, so as to ensure the installation of the gas phase sample above the solution and the complete immersion of the sample in the solution medium. The author made additions and improvements in the paper.
- Figure 1 (page 4 - second "figure 1") - what is the number indicated on the left of the Y axis? line 113: NACE no longer exists, standards have been re-numbered or rebranded.
Answer: Thanks for the reminding of the reviewer. I am sorry for the error in the numbering of the pictures in the article. The author has modified the numbering of the pictures successively and checked the corresponding description of each picture in the article in turn.
(1)The Y-axis represents the corrosion rate of the material, expressed in Vcorr, in mm/a.
(2)The authors modified the numbering of the standards cited in the article according to the AMPP format.
- 172: Unsubstantiated. For confirmation, phase analysis would have to be performed. The mixture likely also includes Fe oxohydroxides.
Answer: Thanks to the reviewer's suggestion. The formation water in the oilfield field almost does not contain dissolved oxygen. Therefore, nitrogen was used to deoxygenate the solution medium before the simulation test to avoid the corrosion of dissolved oxygen. Therefore, there was almost no iron oxide or hydroxide in the corrosion product film. The corrosive gas medium in the test was H2S and CO2, and combined with the element composition detected by EDS, it was inferred that the corrosion products were mainly composed of FeS and FeCO3.
- 190: It is a feasible theory, but unsubstantiated. No proof of peeling is presented.
Answer: Thanks to the reviewer's suggestion. It can be seen that there is a second product film in the area where the local film layer falls off through the corrosion microstructure, indicating that the formation and fall off of the product film is an alternating dynamic balance. References are added to the text.
- Table 3: EDS analysis of whole corrosion layer composition is insufficient. With layer of this thickness, EDS is likely penetrating only top portion and is therefore location-dependant.
Answer: Thanks to the reviewer's suggestion. Before EDS test, the residual solution medium or adhesive impurities on the surface of the sample should be washed with distilled water to ensure that the surface is a clean corrosion product film for observation, and the main element composition of the corrosion product film on the surface of the material should be analyzed under the same multiple and test conditions. Therefore, the test results can be used to compare the element composition of the product film under different test environments, and can be used to infer the types of corresponding corrosion products. This method can also be used to compare the composition of elements in different areas of the same specimen surface, such as the composition of elements inside and outside the pitting pit.
- The structure could be improved. For instance, in chap. 3.2.3 is titled "microscopic corrosion mophology under conditions 2" but it includes corrosion rates, results from condition 1 and other comparison, making it difficult to navigate.
Answer: Thanks to the suggestions of the reviewer. The author has revised and improved the description of this section, especially the sentence referring to the corrosion rate. The corrosion rate is mentioned in this paper only to show that the change of corrosion morphology and the degree of corrosion are related to the corrosion rate.
- 227: EDS does not give the information about the intensification of corrosion of the medium.
Answer: Thanks to the reviewer's suggestion. Combined with EDS analysis results of corrosion product films in liquid phase environment at different temperatures, the author added the detection pattern and element content composition in the paper.
- 229: clarification is required. The solubility of H2S decreases with temperature so the mechanism is likely different to what is described.
Answer: Thank you for your suggestions. In the simulation test, the corrosive gas is coexisting environment of CO2 and H2S. With the increase of temperature, the solubility of the corrosive gas decreases, but with the increase of temperature, the activity of the solution system increases, the electrochemical interaction between the material and the corrosive liquid is strengthened, and the corrosion is intensified.
- Figure 8: these results seem extremely unlikely. In liquid with 186 000 ppm Cl- not to detect any chlorides on the surface is simply impossible. Sample applies for Na and K. Also, quantification of oxygen using EDS is not reliable. It seems that the authors manually selected few elements and ignored all the others.
Answer: Thanks to the reviewer's suggestion. The author also mentioned earlier that before scanning electron microscopy observation and EDS detection, the test sample should be cleaned to remove the residual solution medium and adhesion impurities on the surface, in order to observe the microscopic morphology and elemental composition of the product film. Therefore, the residual Cl, Na and k plasma on the surface of the sample can be easily cleaned off during washing. However, Cl, Na and plasma can also be detected on the relatively thick surface of the product film, because the presence of these elements will affect the content of the component elements of the product film. In the low temperature environment in Fig. 8, the surface film layer of the sample was smooth, and the residual impurities such as salt solution and ions were washed away by the solution. However, Cl ions were detected in the two relatively thick product films in Fig. 8 (c). The author added EDS detection spectra in the paper, and Fig 10 shows the newly added EDS spectra.
- Figure 9: likely breaches copyright, looks like scan or copy from a book.
Answer:Thanks to the reminding of the reviewer. The author modified the original Fig 9 and replaced it with the physical figure of four-point bending loading stress, and change it to Fig 11.
- 239: how was the stress calculated? Again, this is impossible to be true across the cross-section of the bent sample, the inside curve will experience much different stresses than the outside which is the reason why deformation is usually given in this experimental setup.
Answer: Thanks to the reviewer's suggestion. Four-point bending loading method is adopted for stress loading, and the stress calculation and loading is carried out according to GB/T 4157-2017 standard, which has been mentioned in the paper by the author. This method is mainly used to consider the stress sensitivity of the material under the synergistic action of tensile stress and corrosive medium. The inner curved surface of the sample is mainly compressive stress, and the influence of compressive stress is not considered in reference to this standard.
- 293: Just an opinion, but calling 180 °C "ultra-high" seems excessive.
Answer: Thanks to the reviewer's suggestion. The author checked the definition of high temperature and ultra-high temperature in the oil field, and believed that the temperature greater than 180℃ is ultra-high temperature, so the author made modifications in the paper.
- 306: if chlorides penetrate the film, they would have to be found in EDS analysis.
Answer: Based on the corrosion mechanism analysis. Cl- ions can easily penetrate the product film due to their small radius, so it is easy to find the presence of Cl- ions in the product film, especially in the thick product film, which can be found in the EDS detection results in Fig 10 added by the author. The author also stated earlier that the surface of the sample should be washed before the product film morphology analysis and EDS detection, and the residual solution medium and debris should be cleaned to avoid the residual salt solution or the attached impurity ions affecting the morphology observation and the elemental composition of the corrosion product film. Therefore, the residual salt solution and ions on the surface of the rinsed sample are rarely detected unless the product film is loose or thick and cannot be easily rinsed.
- 309: Too broad conclusion to be made based on the experiments performed.
Answer: Thank you for your suggestions. Combined with the experimental results and analysis, the author modified and improved the experimental conclusion.
- English level is mostly ok. Nevertheless, check-up is required, there are words out-of place and non-scientific terms.
Answer: Thank you for your suggestions. The author checked and modified the grammar and language of the whole article, and asked the native English teacher to polish the language, so as to improve the language level of the article.

Reviewer 2 Report
Although, the submitted research work on “Corrosion behavior of tubing in high-salinity formation water environment containing H2S/CO2 in Yingzhong block” appears to be an interesting work. However, there are certain issues with the same. Author(s) could be provided an opportunity to resubmit. Following are initial comments:
1. The usage of English language is not satisfactory at a number of instances. Suggest getting the paper checked thoroughly by a native English speaker prior to submitting the revised version (typo-graphical errors also need to be checked in detail).
2. The discussion section 3.4 presented herein need to be expanded further and be divided into sub-sections as it shall lead to more interpretations. Overall form at present is too clumsy to follow appropriately.
3. Related to comment 2; accordingly conclusions will improve.
4. Likewise, in the Section 2.2; the microscopic part could be separately provided in the form of a sub-section.
5. Stress corrosion cracking is interpreted in the results and analysis section; however there is no corresponding literature or a special mention on the same in the introduction section.
*Author(s) should highlight all the modifications carried out in the paper.
The usage of English language is not satisfactory at a number of instances. Suggest getting the paper checked thoroughly by a native English speaker prior to submitting the revised version (typo-graphical errors also need to be checked in detail).
Author Response
Dear reviewer :
Thank you for the comments on the manuscript, We have revised the manuscript according to your comments. The revised manuscript and the detail responses to the comments of the editor are attached.
- The usage of English language is not satisfactory at a number of instances. Suggest getting the paper checked thoroughly by a native English speaker prior to submitting the revised version (typo-graphical errors also need to be checked in detail).
Answer: Thank you for your suggestions. The author checked and modified the grammar and language of the whole article, and asked the native English teacher to polish the language, so as to improve the language level of the article. In addition, the layout of the article has been carefully checked and checked, especially the number of the picture in the article has been modified and improved.
- The discussion section 3.4 presented herein need to be expanded further and be divided into sub-sections as it shall lead to more interpretations. Overall form at present is too clumsy to follow appropriately.
Answer: Thank you for your suggestions. The author has adjusted the content of section 3.4 and divided it into two sections of corrosion film layer and stress sensitivity for analysis.
- Related to comment 2; accordingly conclusions will improve.
Answer: Thank you for your suggestions. The author has revised and improved the conclusion of the article.
- Likewise, in the Section 2.2; the microscopic part could be separately provided in the form of a sub-section.
Answer: Thank you for your suggestions. The author added section 2.3 to provide a method for detecting and analyzing the microstructure of the product film.
- Stress corrosion cracking is interpreted in the results and analysis section; however there is no corresponding literature or a special mention on the same in the introduction section.
Answer: Thank you for your suggestions. The stress corrosion cracking sensitivity of materials must be analyzed and evaluated in the simulated oil field environment of high temperature and pressure containing H2S. Therefore, the author supplemented and modified the introduction part, and made prominent marks.
- *Author(s) should highlight all the modifications carried out in the paper.
Answer: Thank you very much for your reminding. The author has highlighted all the changes in the text.

Reviewer 3 Report
The topic of studying the corrosiveness of the material in aggressive environments is relevant. Introduction, keywords, abstract, list of references correspond to the stated topic.
The authors confirmed in the conclusions, the known facts that with increasing temperature, there is an increase in the corrosion rate and an increase in metal cracking stresses.
I do not see any scientific novelty in the research, there are no recommendations to reduce the corrosion rate.
Comments
Figure 1. The name of the picture is clear. It is not clear what the black vertical stripes mean.
Add analytical dependencies, specific recommendations, and measures to reduce corrosion rates
Author Response
Dear reviewer :
Thank you for the comments on the manuscript, We have revised the manuscript according to your comments. The revised manuscript and the detail responses to the comments of the editor are attached.
Figure 1. The name of the picture is clear. It is not clear what the black vertical stripes mean.
Add analytical dependencies, specific recommendations, and measures to reduce corrosion rates
Answer: Thanks to the reviewer's suggestion. The black vertical stripes in the picture are the shading of the picture and have no meaning,the author redrew the drawing and removed black vertical stripes. In this paper, the author added the suggestion that appropriate anti-corrosion measures should be taken to reduce the corrosion rate of materials and slow down the corrosion damage of materials. The revisions are highlighted.

Reviewer 4 Report
1. A critical review of existing literature is missing in this manuscript. While the environment is unique, there is literature where tests are performed on similar materials or steels at temperatures higher than 190C or corrosion is performed at very highly saline environments. These should be highlighted and included here as they are factors that go into contributing to corrosion synergistically.
2. Why were the non-stressed and stressed samples having a different sample geometry? Would it not benefit to have the same geometry so that their weight changes and damage morphologies can be compared effectively?
3. Please elaborate on how the sample was polished to mirror finish, how the drying was performed and how the water was removed.
4. What was the technique used to calculate ion composition of the water?
5. Could the authors please explain what the NACE standard dictates and how it was used here for the benefit of the reader?
6. At 180C, do the authors know whether the formation water is in liquid state or vapor state?
7. What is the Y-axis in the weight loss plot? Please elaborate this and the units in the manuscript text. Figure numbers are incorrect and must be changed throughout the manuscript. The text annotations on the plot are misplaced. Were multiple replicates performed for each condition?
8. Section 3.2.1: What is meant by regional corrosion? How do we know condensation is occurring at 180C? It is too high a temperature and even the sample would have achieved equilibrium (during the 168h) with the surrounding and hence offer little opportunity to condense, except maybe at the corner or edges.
9. Sample images taken using different lighting conditions. Please make this uniform throughout the manuscript. No scale bar in any of the images.
10. Why do we not see any Chlorine peaks in the EDS analysis?
11. How exactly does the S accelerate the corrosion? What is the chemical reaction involved here? Corrosion in liquid environment is much worse according to weight loss analysis, but why does the surface look relatively flat and clean in figure 8? What is the pH of the solutions before and after the corrosion process?
12. Why was the test performed at different time compared to corrosion time?
13. How many samples were SCC tested per condition?
14. Different image lighting conditions. How do we know what we see is pitting? Why is the sample pitting? What is the depth of pitting when you say it is immeasurable? What is the microscope's depth resolution? None of this is discussed here.
15. Section 3.4: No evidence is provided for multiple layer thickness! Why is it multiple? what do the layers contain?
16. Why is the role of Chloride ions not discussed here? H+ is discussed, what is the solution pH?
17. How do the authors deconvolute the effect of pH, H+ ion, CO2:H2S ratio, chloride ion content, high temperature and mechanical stress? This is a very complex challenge and the authors have not presented an approach towards understanding this further or do not have enough samples tested/statistics that can help understand their damage behavior further.
The quality of the language has to be improved. Please use technical editing support for this.
Author Response
Dear reviewer :
Thank you for the comments on the manuscript, We have revised the manuscript according to your comments. The revised manuscript and the detail responses to the comments are attached.
1) General comments:
1a) Some grammatical error sees in the article. Please take time to improve the language.
Answer: Thank you for your suggestions. The author checked and modified the grammar and language of the whole article, and asked the native English teacher to polish the language, so as to improve the language level of the article.
2) Keywords and Highlights:
2a) The authors must update keywords in the article. They are not sensible.
Answer: Thank you for your suggestions. The author has modified and improved the whole article, and modified the representative keywords combined with the main research content.
2b) The authors must add Highlights in the article.
Answer: Thank you for your suggestions. Based on the coexisting environment of CO2/H2S and the high temperature environment of highly mineralized formation water, this paper studied and analyzed the corrosion damage degree, stress corrosion cracking sensitivity and the changing trend of corrosion performance of the material under the synergistic action of multiple factors, so as to clarify the adaptability of the material. The author has revised the description of the key content of the paper, and revised the abstract content and conclusion, highlighting the highlights of the paper.
3) Abstract:
3a) The abstract doesn’t have novelty in it. The authors should rewrite the abstract with main novelty in it.
Answer: Thank you for your suggestions. Based on the research purpose and results, the author modified and improved the content of the abstract.
3b) What is the main purpose of the article? The authors should focus on novelty on this section. Please highlight it.
Answer: Thank you for your suggestions. The purpose of this study is mainly to study and analyze the corrosion damage degree, stress corrosion cracking sensitivity and the changing trend of corrosion performance of P110SS material under the synergistic action of multiple factors in the coexisting environment of CO2/H2S and the high temperature environment of highly mineralized formation water, so as to clarify the adaptability of the material to the environment. The author has modified and improved the paper and refined the highlights of the study.
4) Introduction and Literature Review:
4a) The introduction is very short. The authors should extend introduction's length.
Answer: Thank you for your suggestions. According to the research background, the author adds the content of the introduction.
4b) The authors must take a part "Literature Review" from "Introduction".
Answer: Thank you for your suggestions. After consulting a lot of references, the author added some research contents and results of some experts and scholars on the oilfield formation water environment, and compared the environmental media and conditions of the research, clarified the differences between the studies of other scholars and the author, and carried out a comparative analysis.
4c) I strongly suggest, authors compare their work with another researcher (Table 1) (not only put some numerical error without compare the main different between their article and other based on ML algorithms) (major comment).
Answer: Thank you for your suggestions. Table 1 is the ion concentration of solution medium used for simulating high temperature and high pressure test in this paper, so as to clarify the test environmental medium. In the introduction, the author adds the experimental conditions and research results of other researchers to compare the different studies.
5) Methodology:
5a) I strongly suggest authors make one subtitle and write about peculiarity of their methods which used in this topic? (major comment).
Answer: Thank you for your suggestions. The experimental methods including high temperature and high pressure, stress loading method and tissue morphology observation method are supplemented and improved in this paper.
5b) How authors make sure their methods have validated with this data? (major comment).
Answer: Thank you for your suggestions. The high temperature and high pressure method, stress loading method and corrosion rate calculation method used in this paper are all carried out according to the standard, so they are operable. The author adds the corresponding criteria in the paper.
6) Results and Discussions:
6a) It has figures, but technical description for figures is not enough. The authors must describe as well for every figure (major comment).
6b) This section needs to update and need more detail.
6c) The authors should add more detail in this section.
6c) The authors should add more figures to compare in this section
Answer: Thank you for your suggestions. The author refined and supplemented the analysis and discussion of the test results, improved the comparison and analysis of the test data, and updated the detailed description of the test analysis.
7) Conclusion and recommendations:
Conclusion lack of novelty. Please rewrite your conclusion and add some highlight and novelty in it (major comment). Conclusion is so short the authors should extend the material.
Answer: Thank you for your suggestions. The authors have re-summarized the conclusions and detailed the results of the study.
8) Abbreviations:
8a) The authors must update and add the abbreviation in their articles.
Answer: Thank you for your suggestions. The author adds abbreviations to some concepts and technical terms in the text.
9) References:
9a) References should be updated (2022-2023)
Answer: Thank you for your suggestions. According to the expansion and needs of the introduction, the author adds references and updates them.

Reviewer 5 Report
present paper describes a study that aims to " Corrosion behavior of tubing in high-salinity formation water environment containing H2S/CO2 in Yingzhong Block". While the study's topic is of relevance to this journal, the manuscript's content requires major revisions to improve its scientific quality. Upon careful review, I have identified several issues that need to be addressed before the paper can be accepted for publication.
To assist the authors in enhancing the manuscript's quality, I have provided detailed comments outlining the identified flaws. It is essential that the authors address these concerns in a point-by-point manner. If the authors are unable to make the necessary revisions, the article will not meet the standards required for publication in this journal.
In conclusion, this paper has potential, but significant improvements are required before it can be considered for publication. I look forward to reviewing the revised version of the manuscript and thank the authors in advance for their diligent efforts to improve the quality of their work.
================================
1) General comments:
1a) Some grammatical error sees in the article. Please take time to improve the language.
================================
2) Keywords and Highlights:
2a) The authors must update keywords in the article. They are not sensible.
2b) The authors must add Highlights in the article.
================================
3) Abstract:
3a) The abstract doesn’t have novelty in it. The authors should rewrite the abstract with main novelty in it.
3b) What is the main purpose of the article? The authors should focus on novelty on this section. Please highlight it.
================================
4) Introduction and Literature Review:
4a) The introduction is very short. The authors should extend introduction's length.
4b) The authors must take a part "Literature Review" from "Introduction".
4c) I strongly suggest, authors compare their work with another researcher (Table 1) (not only put some numerical error without compare the main different between their article and other based on ML algorithms) (major comment).
================================
5) Methodology:
5a) I strongly suggest authors make one subtitle and write about peculiarity of their methods which used in this topic? (major comment).
5b) How authors make sure their methods have validated with this data? (major comment).
================================
6) Results and Discussions:
6a) It has figures, but technical description for figures is not enough. The authors must describe as well for every figure (major comment).
6b) This section needs to update and need more detail.
6c) The authors should add more detail in this section.
6c) The authors should add more figures to compare in this section
================================
7) Conclusion and recommendations:
7a) Conclusion lack of novelty. Please rewrite your conclusion and add some highlight and novelty in it (major comment).
7b) Conclusion is so short the authors should extend the material.
================================
8) Abbreviations:
8a) The authors must update and add the abbreviation in their articles.
================================
9) References:
9a) References should be updated (2022-2023)
.
Author Response
Dear reviewer :
Thank you for the comments on the manuscript, We have revised the manuscript according to your comments. The revised manuscript and the detail responses to the comments are attached.
1) General comments:
1a) Some grammatical error sees in the article. Please take time to improve the language.
Answer: Thank you for your suggestions. The author checked and modified the grammar and language of the whole article, and asked the native English teacher to polish the language, so as to improve the language level of the article.
2) Keywords and Highlights:
2a) The authors must update keywords in the article. They are not sensible.
Answer: Thank you for your suggestions. The author has modified and improved the whole article, and modified the representative keywords combined with the main research content.
2b) The authors must add Highlights in the article.
Answer: Thank you for your suggestions. Based on the coexisting environment of CO2/H2S and the high temperature environment of highly mineralized formation water, this paper studied and analyzed the corrosion damage degree, stress corrosion cracking sensitivity and the changing trend of corrosion performance of the material under the synergistic action of multiple factors, so as to clarify the adaptability of the material. The author has revised the description of the key content of the paper, and revised the abstract content and conclusion, highlighting the highlights of the paper.
3) Abstract:
3a) The abstract doesn’t have novelty in it. The authors should rewrite the abstract with main novelty in it.
Answer: Thank you for your suggestions. Based on the research purpose and results, the author modified and improved the content of the abstract.
3b) What is the main purpose of the article? The authors should focus on novelty on this section. Please highlight it.
Answer: Thank you for your suggestions. The purpose of this study is mainly to study and analyze the corrosion damage degree, stress corrosion cracking sensitivity and the changing trend of corrosion performance of P110SS material under the synergistic action of multiple factors in the coexisting environment of CO2/H2S and the high temperature environment of highly mineralized formation water, so as to clarify the adaptability of the material to the environment. The author has modified and improved the paper and refined the highlights of the study.
4) Introduction and Literature Review:
4a) The introduction is very short. The authors should extend introduction's length.
Answer: Thank you for your suggestions. According to the research background, the author adds the content of the introduction.
4b) The authors must take a part "Literature Review" from "Introduction".
Answer: Thank you for your suggestions. After consulting a lot of references, the author added some research contents and results of some experts and scholars on the oilfield formation water environment, and compared the environmental media and conditions of the research, clarified the differences between the studies of other scholars and the author, and carried out a comparative analysis.
4c) I strongly suggest, authors compare their work with another researcher (Table 1) (not only put some numerical error without compare the main different between their article and other based on ML algorithms) (major comment).
Answer: Thank you for your suggestions. Table 1 is the ion concentration of solution medium used for simulating high temperature and high pressure test in this paper, so as to clarify the test environmental medium. In the introduction, the author adds the experimental conditions and research results of other researchers to compare the different studies.
5) Methodology:
5a) I strongly suggest authors make one subtitle and write about peculiarity of their methods which used in this topic? (major comment).
Answer: Thank you for your suggestions. The experimental methods including high temperature and high pressure, stress loading method and tissue morphology observation method are supplemented and improved in this paper.
5b) How authors make sure their methods have validated with this data? (major comment).
Answer: Thank you for your suggestions. The high temperature and high pressure method, stress loading method and corrosion rate calculation method used in this paper are all carried out according to the standard, so they are operable. The author adds the corresponding criteria in the paper,
6) Results and Discussions:
6a) It has figures, but technical description for figures is not enough. The authors must describe as well for every figure (major comment).
6b) This section needs to update and need more detail.
6c) The authors should add more detail in this section.
6c) The authors should add more figures to compare in this section
Answer: Thank you for your suggestions. The author refined and supplemented the analysis and discussion of the test results, improved the comparison and analysis of the test data, and updated the detailed description of the test analysis.
7) Conclusion and recommendations:
Conclusion lack of novelty. Please rewrite your conclusion and add some highlight and novelty in it (major comment). Conclusion is so short the authors should extend the material.
Answer: Thank you for your suggestions. The authors have re-summarized the conclusions and detailed the results of the study.
8) Abbreviations:
8a) The authors must update and add the abbreviation in their articles.
Answer: Thank you for your suggestions. The author adds abbreviations to some concepts and technical terms in the text.
9) References:
9a) References should be updated (2022-2023)
Answer: Thank you for your suggestions. According to the expansion and needs of the introduction, the author adds references and updates them.

Round 2
Reviewer 1 Report
If possible, please avoid using description of conditions as "condition 1" and "condition 2". It does not save any space compared to "high H2S" or "low H2S" and makes the reader scroll up unnecessarily.
The language quality could still be improved. There is a number of words out of context (e.g. "vitality of chlorides") and typographical errors.
Author Response
Dear reviewer :
Thank you for the comments on the manuscript, We have revised the manuscript according to your comments,and the detail responses to the comments of the editor are attached.
- If possible, please avoid using description of conditions as "condition 1" and "condition 2". It does not save any space compared to "high H2S" or "low H2S" and makes the reader scroll up unnecessarily.
Answer: Thanks to the reviewer's suggestion. The author revised the statement of test conditions in this paper, and excluded conditions 1 and 2 from the description of test conditions.
- The language quality could still be improved. There is a number of words out of context (e.g. "vitality of chlorides") and typographical errors.
Answer: Thanks to the reviewer's suggestion. The author has revised and improved the entire article, rectifying inappropriate expressions, typographical errors, and grammar mistakes.

Reviewer 2 Report
Revised manuscript is satisfactory.
Author Response
Dear reviewer :
Thank you very much for your support and recognition of our work. We hope to have better communication with you in the future. Thank you.

Reviewer 3 Report
comments are fixed, accept for publicationAuthor Response
Dear reviewer :
Thank you very much for your support and recognition of our work. We hope to have better communication with you in the future. Thank you.

Reviewer 4 Report
I still do not the changes I recommended to the manuscript such as information regarding weight changes, changes to sample images (no scale bar), clearer understanding of microscopy images, rationale for different sample configs, and a thorough analysis of why certain factors are more important than others.
Significant improvement still needed. Sentences are way too long, and not constructed aptly.
Author Response
Dear reviewer :
Answer: Thanks to the reviewer's suggestion. Based on your suggestion, the author has made some additions to the testing method, particularly regarding weightlessness and changes in weight. The author included a scale in the picture, supplemented information about the main factors that cause corrosion, and added references. Additionally, the author revised and improved grammar and sentence structure throughout the article. Many thanks to the judges for their guidance and suggestions.

Reviewer 5 Report
I am glad that the authors effectively addressed my concerns and challenges in their research work. The authors' ability to provide timely and satisfactory responses to my queries reflects their strong commitment to adhering to scientific principles and conducting reliable research. This dedication benefits the scientific community and enhances our understanding of the subject matter. Therefore, based on the authors' satisfactory response, I find this version of the article to be acceptable.
Author Response

(The authors gave the same response as above.)
